# RESERVOIR TRANSFORMERS

## ABSTRACT

We demonstrate that transformers obtain impressive performance even when some of the layers are randomly initialized and never updated. Inspired by old and well-established ideas in machine learning, we explore a variety of non-linear "reservoir" layers interspersed with regular transformer layers, and show improvements in wall-clock compute time until convergence, as well as overall performance, on various machine translation and (masked) language modelling tasks.

## 1  INTRODUCTION

Transformers (Vaswani et al., 2017) have dominated natural language processing (NLP) in recent years, from large scale machine translation (Ott et al., 2018) to pre-trained (masked) language modeling (Devlin et al., 2018; Radford et al., 2018), and are becoming more popular in other fields as well, from reinforcement learning (Vinyals et al., 2019) to speech recognition (Baevski et al., 2019) and computer vision (Carion et al., 2020). Their success is enabled in part by ever increasing computational demands, which has naturally led to an increased interest in improving their efficiency. Scalability gains in transformers could facilitate bigger, deeper networks with longer contexts (Kitaev et al., 2020; Wang et al., 2020; Beltagy et al., 2020; Kaplan et al., 2020; Tay et al., 2020b). Conversely, improved efficiency could reduce environmental costs (Strubell et al., 2019) and hopefully help democratize the technology.

In this work, we explore a simple question: if some layers of the transformer are kept frozen—i.e., never updated after random initialization—can we match the performance of fully learned transformers, while being more efficient? Surprisingly, the answer is resoundingly yes; and what is more, we find that freezing layers may actually *improve* performance.

Beyond desirable efficiency gains, random layers are interesting for several additional reasons. Fixed randomly initialized networks (Gallicchio & Scardapane, 2020) converge to Gaussian processes in the limit of infinite width (Daniely et al., 2016), have intriguing interpretations in metric learning (Rosenfeld & Tsotsos, 2019; Giryes et al., 2016), and have been shown to provide excellent "priors" either for subsequent learning (Ulyanov et al., 2018) or pruning (Frankle & Carbin, 2018). Fixed layers allow for efficient low-cost hardware implementations (Schrauwen et al., 2007) and can be characterized using only a random number generator and its seed, which might have repercussions in distributed training and enables highly efficient deployment to edge devices. The strong performance of networks with fixed layers also sheds new light on the inner workings of BERT (Devlin et al., 2018), and layer-wise interpretations of such models (Rogers et al., 2020; Tenney et al., 2019). It appears that "not all layers are created equal" (Zhang et al., 2019) is true to such an extent that some layers can simply remain random and fixed.

These ideas have a long history in machine learning. By Cover's theorem (Cover, 1965), any high-dimensional non-linear transformation is more likely to be linearly separable than its lower-or-equal-dimensional input space. By Johnson-Lindenstrauss (Johnson & Lindenstrauss, 1984), random projections distort Euclidean distances very little under mild assumptions, which is useful e.g. for dimensionality reduction and random indexing (Sahlgren, 2005). Fixed random layers in neural networks pre-date deep learning by far (Gamba et al., 1961; Baum, 1988). Indeed, random kernel methods have been an impactful idea in machine learning (Rahimi & Recht, 2008; 2009).

One way to think of such layers is as "reservoirs" (Lukoševičius & Jaeger, 2009), where a highly non-linear high-dimensional black box representation is provided to a lightweight "readout" network, as in echo state networks (Jaeger, 2003) and liquid state machines (Maass et al., 2002). The

benefit of such an approach is that the reservoir has fixed parameters and is computationally efficient, as it can be pre-computed and does not (necessarily) require backpropagation.

In NLP, Wieting & Kiela (2019) showed that random sentence encoders present a strong baseline for text classification, with subsequent work showing applications in a variety of NLP tasks (Enguehard et al., 2019; Garg et al., 2020; Pilault et al., 2020). To our knowledge, this work is the first to examine this phenomenon in transformers, and the first to recursively alternate reservoirs with subsequent transformer layers acting as readout functions. We introduce "reservoir transformers", wherein fixed random reservoir layers are interspersed with regular updateable transformer layers. The goal of this work is not necessarily to set a new state of the art, but to put our understanding of transformer models on a more solid footing by providing empirical evidence of their capabilities even when some of their parameters are fixed. Our contributions are as follows:

- We introduce a new *area under the convergence curve* metric for measuring performance-efficiency trade-offs, and show that replacing regular transformer layers with reservoir layers leads to better results on that metric.

- We show that the addition of reservoir layers in fact leads to improved test set generalization on a variety of tasks in a variety of settings.

- We show that pre-trained masked language modelling architectures like BERT and RoBERTa (Liu et al., 2019) can benefit from having some of their layers frozen, both during pre-training as well as when fine-tuning on downstream tasks.

- In addition, we experiment with different types of reservoir layers, including convolutional and recurrent neural network-based ones. We also show empirical evidence that the backward pass can be entirely skipped by approximating top-layer gradients using an approach we call *backskipping*, with a relatively small sacrifice in performance.

## 2 APPROACH

This paper is based on a very simple idea. Neural networks are trained via backpropagation, which involves consecutive steps of matrix addition and multiplication, i.e.,

$$\theta_{t+1} \leftarrow \theta_t - \eta \frac{\partial J}{\partial \theta_t}; \frac{\partial J}{\partial \theta_t} = \frac{\partial J}{\partial L_n} \frac{\partial L_n}{\partial L_{n-1}} \cdots \frac{\partial L_1}{\partial L_0} \frac{\partial L_0}{\partial x} \tag{1}$$

for some objective $J$, parameterization $\theta$ and learning rate $\eta$, with the gradient computed via the chain rule, where $L_i$ is the $i$-th layer of the neural network and $x$ is the input. Let $L = \text{Transformer}(X)$ be a single layer in a Transformer network (Vaswani et al., 2017), i.e.,

$$H = \text{MultiHeadSelfAttn}(\text{LayerNorm}(X)) + X$$
$$L = \text{FFN}(\text{LayerNorm}(H)) + H \tag{2}$$

Now, during every "backward pass", we compute the Jacobian for parameters $\theta^L$ at layer $L$, which are used to update the parameters of $L$, $\theta_t^L$, as well as to compute the next layer's Jacobian, thus back-propagating the gradients. In this work however, for some of the layers, we still backpropagate through them to compute gradients for earlier layers, *but we never update their parameters*. As a result, these layers stay fixed at their random initialization, saving computational resources.

### 2.1 BACKGROUND

Naturally, never updating some of the parameters is computationally more efficient, as some matrix addition operations can be skipped in the backward pass, but why is this not detrimental to the performance of the network?

In the early days of neural networks, the bottom layers were often kept fixed as "associators" (Block, 1962), or what Minsky & Papert (2017) called the Gamba perceptron (Gamba et al., 1961; Borsellino & Gamba, 1961). Fixed random networks (Baum, 1988; Schmidt et al., 1992; Pao et al., 1994) have

been explored from many angles, including as "random kitchen sink" kernel machines (Rahimi & Recht, 2008; 2009), "extreme learning machines" (Huang et al., 2006) and reservoir computing (Jaeger, 2003; Maass et al., 2002; Lukoševičius & Jaeger, 2009). In reservoir computing, input data are represented through fixed random high-dimensional non-linear representations, called "reservoirs", which are followed by a regular (often but not necessarily linear) "readout" network to make the final classification decision.

The theoretical justification for these approaches lies in two well-known results in machine learning: Cover's theorem (Cover, 1965) on the separability of patterns states that high-dimensional non-linear transformations are more likely to be linearly separable; and the Johnson-Lindenstrauss lemma (Johnson & Lindenstrauss, 1984) shows that random projections distort Euclidean distances very little under mild assumptions.

Practically, random layers can be seen as a cheap way to increase network depth. There are interesting advantages to this approach. Fixed layers are known to have particularly low-cost hardware requirements and can be easily implemented on high-bandwidth FPGAs with low power consumption (Hadaeghi et al., 2017; Tanaka et al., 2019), or on optical devices (Hicke et al., 2013). This might yield interesting possibilities for training in a distributed fashion across multiple devices, as well as for neurmorphic hardware (Neftci et al., 2017). This approach also facilitates lower-latency deployment of neural networks to edge devices, since weights can be shared simply by sending the seed number, assuming the random number generator is known on both ends.

## 2.2 RESERVOIR TRANSFORMERS

This work explores inserting random non-linear transformations, or what we call reservoir layers, into transformer networks. Specifically, we experiment with a variety of reservoir layers:

- Transformer Reservoir: The standard transformer layer as described above, but with all parameters fixed after initialization, including the self-attention module.
- FFN Reservoir: A transformer-style fixed feed-forward layer without any self-attention, i.e., FFN(LayerNorm(Previous_layer)) + Previous_layer.
- BiGRU Reservoir: A fixed bidirectional Gated Recurrent Unit (Cho et al., 2014) layer, which is closer in spirit to previous work on reservoir computing, most of which builds on recurrent neural network architectures.
- CNN Reservoir: A fixed Convolutional Neural Network (LeCun et al., 1998) layer, specifically light dynamical convolution layers (Wu et al., 2019), which are known to be competitive with transformers in sequence-to-sequence tasks.

We find that all these approaches work well, to a certain extent. For clarity, we focus primarily on the first two reservoir layers, but include a broader comparison in Appendix A.

In each case, contrary to traditional reservoir computing, our reservoir layers are interspersed throughout a regular transformer network, or what we call a reservoir transformer. A good justification for this approach is that while random projections are not learned and might introduce noise, subsequent normal transformer "readout" layers might allow us to recover from any adverse effects of randomness. For example, previous work has shown that ResNets, with all of their parameters fixed except for the scale and shift parameters of batch normalization, can still achieve high performance, simply by scaling and shifting random features (Frankle et al., 2020). Adding noise to the parameters of neural networks is also known to help convergence and generalization (Jim et al., 1995; 1996; Gulcehre et al., 2016; Noh et al., 2017).

## 3 EVALUATION

We evaluate the proposed approach on a variety of well-known tasks in natural language processing, namely: machine translation, language modelling and masked language model pre-training.

In this work, we are not necessarily interested in obtaining the state of the art on any task or even in improving overall task performance via this method. The main objective is to examine efficiency, i.e. the relationship between compute time and task performance. This is closely related

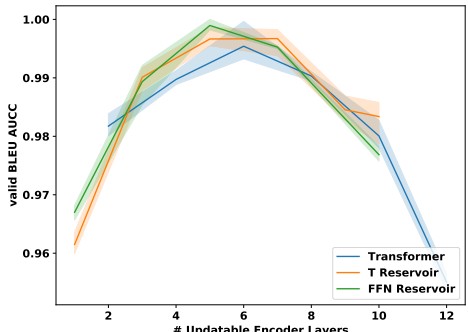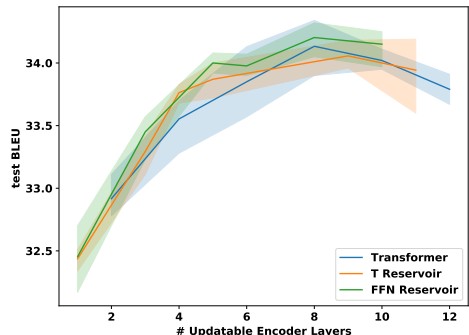

Figure 1: Validation BLEU AUCC and test BLEU for IWSLT (high is good). Comparison of regular transformer and reservoir transformer with FFN or Transformer reservoir layers added.

to efforts in Green AI, which are concerned with the trade-offs between compute, data, and performance (Schwartz et al., 2019). We propose a new metric for our purposes, the *area under the convergence curve* (AUCC): similarly to how the area under the receiver operating characteristic (Bradley, 1997, AUC-ROC) measures a classifier's performance independent of the classification threshold, AUCC measures a model's performance independent of the specific compute budget. Specifically, AUCC is computed as follows:

$$\int_{t=0}^{\hat{T}} \sum_{x,y \in \mathcal{D}} g_t(f(x), y) \tag{3}$$

where $f$ is the network and $g$ is the evaluation metric, measured until convergence time $\hat{T}$, which is the maximum convergence time of all models included in the comparison. Note that time here is wall-clock time, not iterations. By convergence, we mean that validation performance has stopped improving, and hence the convergence curve whose area we measure plots the desired metric over time. Runs are averaged over multiple seeds and reported with standard deviation. We normalize raw AUCC scores by their maximum score to ensure a more easily interpretable $[0 - 1]$ range.

One potential downside of this approach is that the AUCC metric could lead to higher scores for a model that converges quickly but to ultimately worse performance, if measured in a small window. We account for this by making sure that $\hat{T}$ is set sufficiently high. We include the raw validation curves in the appendix and also report test set generalization in each experiment.

## 3.1 EXPERIMENTAL SETTINGS AND IMPLEMENTATION DETAILS

We evaluate on IWSLT de-en (Cettolo et al., 2015) and WMT en-de (Bojar et al., 2014) for machine translation; enwik8 (LLC, 2009) for language modelling; and experiment with RoBERTa (Liu et al., 2019) in our pretraining experiments. For IWSLT, we follow the pre-processing steps in Edunov et al. (2018). The train/val/test split is 129k/10k/6.8k sentences. For WMT, we follow the pre-processing steps in Ott et al. (2018). The train/val/test split is 4.5M/16.5k/3k sentences. For enwik8, we follow the pre-processing steps in Dai et al. (2019). The train/val/test split is 1M/54k/56k sentences. For RoBERTa pretraining, we follow the pre-processing steps in Liu et al. (2019).

We use 8 Volta V100 GPUs for WMT and enwik8, 32 V100 GPUs for RoBERTa and a single V100 for IWSLT. The hyperparameters for IWSLT14 and WMT16 were set to the best-performing values from Ott et al. (2018) and Kasai et al. (2020) respectively. The enwik8 experiment settings followed Bachlechner et al. (2020) and the RoBERTa experiments followed Liu et al. (2019). All experiments were conducted using fairseq (Ott et al., 2019). Our code and experimental settings will be made open source at [ANONYMIZED-GITHUB-URL].

| Model | # Layers | Frozen | Max BLEU | Train time until max (in hours) | Ratio | # Params Trainable (Total) | Train Time each epoch (in seconds) |
|---|---|---|---|---|---|---|---|
| Transformer | 6 | 0 | 34.52 ± 0.07 | 2.548 ± 0.06 | 1 | 26.8M | 122.73 ± 1.16 |
| | 8 | 0 | 34.59 ± 0.11 | 2.557 ± 0.05 | 1 | 31.1M | 142.28 ± 1.87 |
| | 10 | 0 | 34.56 ± 0.05 | 3.173 ± 0.04 | 1 | 35.3M | 161.66 ± 1.54 |
| | 12 | 0 | 34.29 ± 0.12 | 3.521 ± 0.09 | 1 | 39.5M | 172.45 ± 1.98 |
| T Reservoir | 6 | 2 | 34.37 ± 0.12 | 2.422 ± 0.03 | 0.95 | 22.6M (26.8M) | 120.59 ± 1.32 |
| | 8 | 2 | 34.80 ± 0.07 | 2.450 ± 0.06 | 0.96 | 26.8M (31.1M) | 134.49 ± 1.76 |
| | 10 | 2 | 34.70 ± 0.03 | 2.831 ± 0.05 | 0.89 | 31.1M (35.3M) | 144.42 ± 1.98 |
| | 12 | 2 | 34.78 ± 0.04 | 3.476 ± 0.04 | 0.98 | 35.3M (39.5M) | 159.43 ± 1.67 |
| FFN Reservoir | 6 | 2 | 34.43 ± 0.15 | 2.120 ± 0.04 | 0.83 | 22.6M (25.8M) | 107.71 ± 1.73 |
| | 8 | 2 | 34.56 ± 0.16 | 2.203 ± 0.06 | 0.86 | 26.8M (29.1M) | 120.07 ± 1.65 |
| | 10 | 2 | 34.66 ± 0.02 | 2.493 ± 0.05 | 0.79 | 31.1M (33.3M) | 130.11 ± 1.43 |
| | 12 | 2 | 34.76 ± 0.03 | 3.241 ± 0.04 | 0.92 | 35.3M (37.5M) | 156.32 ± 1.87 |
| LayerDrop | 6 | 2 | 34.59 ± 0.15 | 2.364 ± 0.08 | 0.92 | 22.6M (26.8M) | 119.30 ± 1.36 |
| | 8 | 2 | 34.58 ± 0.16 | 2.554 ± 0.05 | 0.99 | 26.8M (31.1M) | 138.62 ± 1.44 |
| | 10 | 2 | 34.57 ± 0.07 | 3.404 ± 0.06 | 1.07 | 31.1M (35.3M) | 140.88 ± 1.62 |
| | 12 | 2 | 33.65 ± 0.24 | 3.251 ± 0.04 | 0.92 | 35.3M (39.5M) | 160.85 ± 1.49 |

Table 1: Wall-clock time (averaged over multiple runs) saved for IWSLT for different model types and encoder depths. Max BLEU is for validation. Number of layers is for encoder, decoder depth is kept fixed at 2. Ratio is computed compared to comparable number of layers in the normal case.

All the experiments in this paper were run with 3 random seeds and the mean and standard deviation are reported. For the relatively small IWSLT, the $\hat{T}$ value in the AUCC metric was set to 4 hours. For WMT, which is larger, we set it to 20 hours. For enwiki8, it was 30 hours; and for the RoBERTa pre-training experiments, it was set to 60 hours.

The projection weights in random layers were initialized using orthogonal initialization (Saxe et al., 2013), which makes sense since random orthogonal projections should be most information-preserving, and which was found to work well empirically for initializing fixed random representations in previous work (Wieting & Kiela, 2019). Biases and layer norm parameters were initialized using their respective PyTorch defaults (based on Xavier init; Glorot & Bengio, 2010).

We intersperse reservoir layers in alternating fashion starting from the middle. Specifically, we alternate one reservoir layer with one transformer layer, and place the alternating block in the middle. For example: a 7-layer encoder LLLLLLL in which we replace three layers with reservoirs becomes LRLRLRL, and with two becomes LLRLRLL. See Appendix C for a study comparing this strategy to alternative approaches (e.g., freezing in the bottom, middle or top).

## 4 EXPERIMENTS

In what follows, we first show our main result: reservoir transformers often have better AUCC metrics, less training time per epoch, less convergence time until the best validation performance is achieved, and even improved test set generalization metrics, on a variety of tasks. As a strong baseline method, we compare to LayerDrop (Fan et al., 2019). LayerDrop can also be seen as a method that dynamically bypasses parts of the computation during Transformer training in an attempt to improve efficiency, and is a suitable comparison to examine our methods.. We also examine whether we can minimize the expectation over the gradients of upper layers in the transformer network such that we do not have to pass the true gradients through the reservoir for further efficiency.

### 4.1 MACHINE TRANSLATION

Machine translation (MT) is one of the core tasks of NLP. We demonstrate on two well-known MT datasets, IWSLT'14 German-English and WMT'16 English-German, that reservoir transformers obtain a better AUCC. For the raw validation plots over time that were used to calculate the AUCC, please refer to Appendix F.

Following Kasai et al. (2020), the architecture of the network is an N-layer reservoir transformer encoder, followed by a regular shallow one- or two-layer decoder. This design choice has been shown to lead to very good speed and efficiency trade-offs, and serves as a good baseline for our

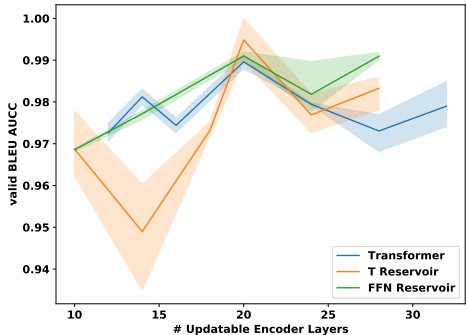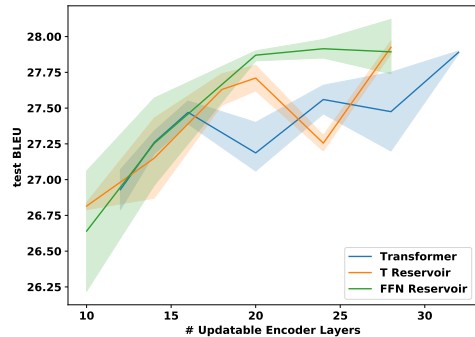

Figure 2: Validation BLEU AUCC and test BLEU for WMT (high is good). Comparison of regular transformer and reservoir transformer with FFN or Transformer reservoir layers added.

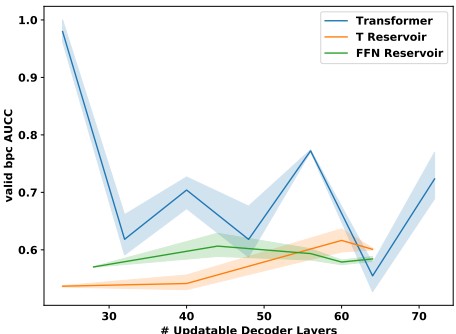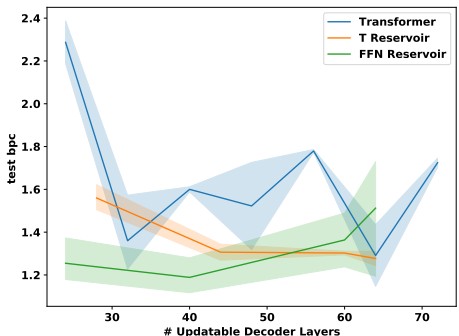

Figure 3: Validation BPC AUCC and test BPC on the enwik8 language modelling task (low is good). Comparison of regular and reservoir transformers for varying depths.

experiments. Moreover, shallow decoders make it easier to decide where to place reservoir layers (in the encoder) and makes it more straightforward to identify where performance gains come from.

Figure 1 shows the results for IWSLT. On the y-axis we show validation AUCC for the BLEU metric; on the x-axis we show the number of updatable layers in the encoder. The performance of a regular transformer encoder with 6 layers and a reservoir transformer encoder with 6 layers plus N additional reservoir layers are plotted for the same x-axis value to show the total number of *updated* layers. Plots for the *total* number of layers (updatable plus not-updatable, so essentially shifted versions) are shown in Appendix E. Table 1 shows the time it took to achieve the maximum validation BLEU score and how that relates to the regular transformer, demonstrating that reservoir transformers consistently converge faster in terms of wall-clock time, up to 22% as much with the same number of updateable layers. We save as much as 27% time until convergence a 24 layer model on WMT, as shown in Table 3. One other noticeable point is that we can see that the T Reservoir achieves similar performance to LayerDrop on IWSLT and WMT in terms of wall-clock per epoch and wall-clock time to the best performance. However, on both tasks, FFN Reservoir performs much better than LayerDrop in terms of efficiency per epoch and achieves better/similar performance in less time in each case. As a point of reference, a half hour gain on IWSLT translates to a gain of several days in the training of bigger transformer models like GPT-3 (Brown et al., 2020).

We observe that reservoir transformers consistently perform better than, or are competitive to, regular transformers, both in terms of validation BLEU AUCC as well as test time BLEU, for all examined encoder depths.

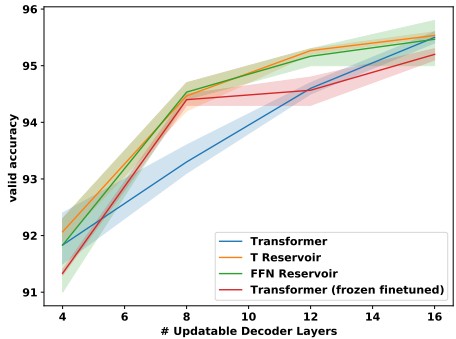 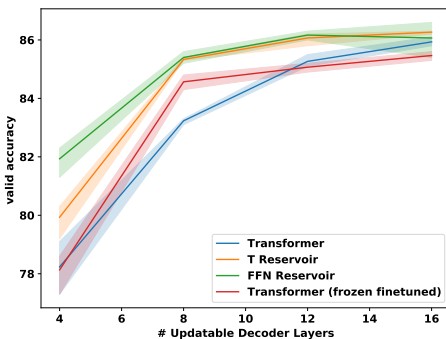

Figure 4: Downstream RoBERTa performance on SST-2 (left) and MultiNLI-matched (right).

Figure 2 shows a similar trend for WMT. WMT is much larger and requires a much deeper encoder, as illustrated by the fact that a certain minimum depth is required for reservoir transformers to achieve a comparable validation AUCC. At test time, reservoir transformers outperform regular transformers for almost all encoder depths. The FFN reservoir transformer seems to work best in both cases, which is surprising because it does not have any self-attention component at all. This finding shows that self-attention, or the mechanism to summarize context information, should be learned if present. Once the context features have been gathered, a random projection via a fixed FFN module appears to be beneficial, at least for MT.

### 4.2 Language Modelling

To examine whether the same findings hold for other tasks, we evaluate on the enwiki8 (LLC, 2009) language modelling task. We examine the BPC (bits per character) rate for a variety of network depths (since the task is language modelling, these layers are in the decoder). The results show that we obtain consistently better BPC for lower depths, except for the 64-layer regular transformer, which appears to be particularly optimal for this task. We observe similar trends during test time.

### 4.3 Masked Language Model Pretraining

We train RoBERTa (Liu et al., 2019) models from scratch at a variety of depths, both in the normal and reservoir setting. We find that these networks show minor differences in their best perplexity and similar AUCC perplexity (see Appendix D). We then examine the performance of these models when fine-tuned on downstream tasks, specifically the well known SST-2 (Socher et al., 2013) and MultiNLI[1] (Williams et al., 2017) tasks. When fine-tuning the reservoir models, we keep the reservoir layers fixed (including them in fine-tuning did not work very well, see Appendix D).

Figure 4 shows the results of fine-tuning. We observe that the reservoir transformer outperforms normal RoBERTa at all depths in both tasks. At lower depth, the improvements are substantial. As a sanity check, we also experiment with freezing some of the layers in normal RoBERTa during fine-tuning (Transformer frozen finetuned) and show that this helps a little but is still outperformed by the reservoir transformer.

These findings suggest that you can train a RoBERTa model without updating all of the layers, achieve similar perplexity at a similar computational cost, but with better downstream performance. The fact that some layers can be kept random and entirely fixed during training, without sacrificing any performance, raises intriguing questions for "BERTology" (Rogers et al., 2020) and for the study of what different layers in transformers learn.

---

[1]We report results for MultiNLI-Matched.

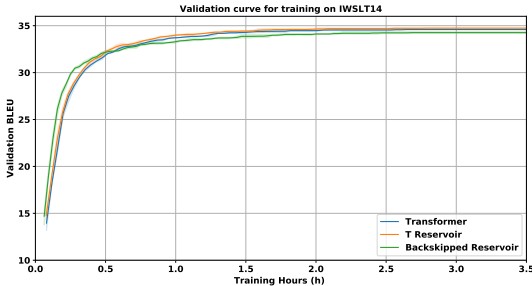

Figure 5: IWSLT comparison of normal v frozen v backskipped

## 4.4 BACKSKIPPING

With the reservoir transformers as described above, we obtain better efficiency by skipping the "gradient application" matrix addition step in some of the layers (i.e., updating the weights). One step further would be to investigate skipping the entire backward pass for reservoirs altogether, which would save us from having to do the much more expensive matrix multiplication for these layers that is required for the propagation of gradients. We report on preliminary experiments where in the backward pass we replace the gradients for the layer $L_i$ *going into* the reservoir $L_{i+1}$ with a noisy estimate (Jaderberg et al., 2017; Czarnecki et al., 2017). Promisingly, Oktay et al. (2020) recently asked "why spend resources on exact gradients when we're going to use stochastic optimization?" and show that you can do randomized auto-differentiation quite successfully.

Here, rather than minimizing the actual gradients $\frac{\partial L_i}{\partial \theta^{L_i}}$, we minimize their expectation and train via continuous-action REINFORCE (Williams, 1992). That is, $L_i$ becomes a policy $\pi_a$: $s \rightarrow \mu$ where we sample actions $a \sim \mathcal{N}(\mu, 1)$. We train to minimize the gradient prediction loss via MSE, i.e., $\frac{1}{n} \sum_{i=0}^{n} (R^i - V^i(a))^2$, and the REINFORCE loss $\mathbb{E}_a \left[ \log(a) \left( R - V(a) \right) \right]$, where the value network $V$ acts as the baseline. $R$ is defined as the mean of the gradients of the top layer $L_{i+2}$, with the sign flipped. Thus, simply put, we train to minimize the expectation of the true gradients at the layer directly following the reservoir. We employ an annealing scheme where we first train the value network and propagate the true gradients during warmup. Afterwards, we anneal the probability of backskipping rather than performing a true backward pass (multiplying the probability by 0.99 every iteration until we only backskip). We experimented with setting $R$ to the negation of the total loss as well but found the current reward to work better. We call this approach *backskipping*.

Figure 5 shows the results as validation BLEU over time. We observe that this approach helps especially during the earlier stages of training. Although it does not match the performance of the approach with true gradients quite yet, it actually performs competitively. Backskipping looks promising as an approach to further reduce computational costs, and would be even more efficient from a hardware perspective since the circuitry for such layers (which do not need to propagate gradients) can effectively be hardwired entirely.

## 5 RELATED WORK

Recent work has shown that modern NLP models are able to function with different numbers of layers for different examples (Elbayad et al., 2019; Fan et al., 2019); that different layers specialize for different purposes (Zhang et al., 2019); that layers can be compressed (Li et al., 2020); and, that layers can be reordered (Press et al., 2019). There is a growing body of work in efficient self-attention networks (Tay et al., 2020b), such as linear attention (Wang et al., 2020), on how to process long context information (Beltagy et al., 2020) and on approximations to make transformers more scalable (Kitaev et al., 2020; Katharopoulos et al., 2020). BigBIRD (Zaheer et al., 2020) provides random keys as additional inputs to its attention mechanism. Locality sensitive hashing (LSH) as employed e.g. in Reformer (Kitaev et al., 2020) utilizes a fixed random projection. Performer (Choromanski et al., 2020) computes the transformer's multi-head attention weights as a fixed orthogonal random projection. Closely related to this work, Tay et al. (2020a) showed

that randomized alignment matrices in their "Synthesizer" architecture are sufficient for many NLP tasks. While these works focus on random attention, we show that *entire* layers can be random and fixed. We also show that entire layers can be replaced by fixed random projections that do not have any attention whatsoever.

Beyond transformers, random features have been extensively explored. Examples of this include FreezeOut (Brock et al., 2017), deep reservoir computing networks (Scardapane & Wang, 2017; Gallicchio & Micheli, 2017), as well as applications in domains as varied as text classification (Conneau et al., 2017; Zhang & Bowman, 2018; Wieting & Kiela, 2019) or music classification (Pons & Serra, 2019). It is well known that randomly initialized networks can display impressive performance on their own (Ulyanov et al., 2018; Rosenfeld & Tsotsos, 2019; Ramanujan et al., 2020), which underlies, for example, the recently popularized lottery ticket hypothesis (Frankle & Carbin, 2018; Zhou et al., 2019). We know that learning deep overparameterized networks appears to help in general (Li & Liang, 2018; Du et al., 2019). Our method represents an easy and cheap way to add both depth and parameters to transformer networks.

## 6 Conclusion

This work demonstrated that state-of-the-art transformer architectures can be trained without updating all of the layers. This complements a long history in machine learning of harnessing the power of random features. In most cases, "reservoir transformers" achieve better performance-efficiency trade-offs as measured by our newly introduced AUCC metric, and better test set generalization, on a variety of tasks and in a variety of settings. Future work includes further investigating hybrid networks and backskipping architectures, as well as utilizing pruning strategies at inference time, in order to try to obtain even better performance/efficiency trade-offs.

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

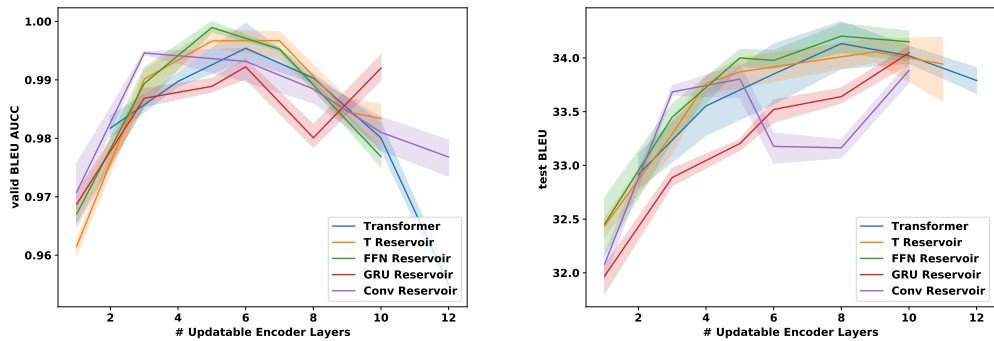

Figure 6: IWSLT comparison of different hybrid architectures with different reservoir layers.

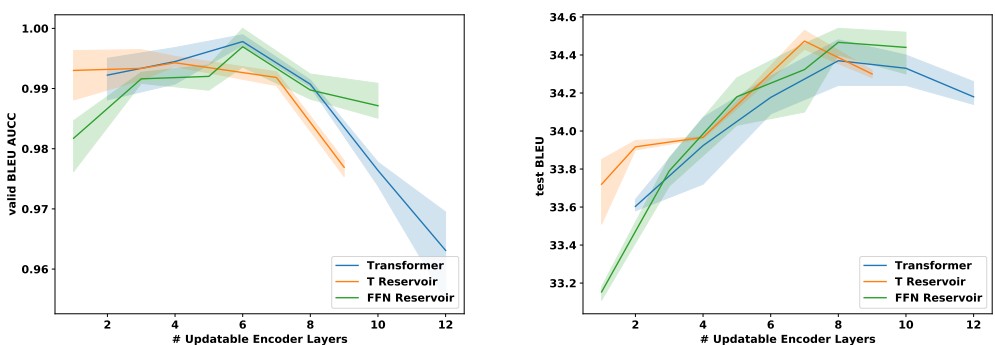

Figure 7: IWSLT validation AUCC and test BLEU with 6-layer decoder.

## A  HYBRID NETWORKS AND NON-TRANSFORMER RESERVOIRS

We investigate whether reservoir layers need to be transformer-based (or transformers-without-attention, i.e., FFN). We examine two different alternatives: bidirectional Gated Recurrent Units (Cho et al., 2014) and Convolutional Neural Networks (LeCun et al., 1998; Kim, 2014), specifically light dynamical convolutions (Wu et al., 2019). Figure 6 shows the results for these hybrids: depending on the setting, they may obtain a better AUCC than the regular transformer, but this is less consistent than with the other reservoir layers, most likely because these layers have different computational properties. It's possible that these hybrids simply require further tuning, as we found e.g. up-projecting to help for BiGRUs, but studying this is outside of the scope of the current work.

## B  DEEP DECODERS

We show that the same results hold for a 6-layer decoder on IWSLT (although less pronounced for AUCC, probably because the decoder is computationally heavier). See Figure 7 and Table 2.

## C  FREEZING STRATEGY

We explored different strategies for the placement of reservoir layers and found the "alternating" strategy reported in the main body of the paper to work best. Generally, we found repetitive application of reservoirs to yield diminishing returns, as might be expected. See Figure 8.

| Model | # Layers | Frozen | Max BLEU | Train time until max (in hours) | Ratio | # Params Trainable (Total) | Train Time each epoch (in seconds) |
|---|---|---|---|---|---|---|---|
| Transformer | 6 | 0 | $34.97 \pm 0.05$ | $1.984 \pm 0.02$ | 1 | 39.5M | $177.84 \pm 2.98$ |
| | 8 | 0 | $34.99 \pm 0.08$ | $2.161 \pm 0.03$ | 1 | 43.7M | $206.59 \pm 3.47$ |
| | 10 | 0 | $34.98 \pm 0.04$ | $2.345 \pm 0.02$ | 1 | 47.9M | $236.72 \pm 3.52$ |
| | 12 | 0 | $34.78 \pm 0.11$ | $2.535 \pm 0.05$ | 1 | 52.0M | $265.90 \pm 4.97$ |
| T Reservoir | 6 | 2 | $34.73 \pm 0.11$ | $1.838 \pm 0.01$ | 0.92 | 35.3M (39.5M) | $166.11 \pm 2.21$ |
| | 8 | 2 | $35.07 \pm 0.05$ | $1.912 \pm 0.03$ | 0.88 | 39.5M (43.7M) | $190.08 \pm 3.73$ |
| | 10 | 2 | $35.02 \pm 0.01$ | $1.970 \pm 0.04$ | 0.84 | 43.7M (47.9M) | $204.42 \pm 2.89$ |
| | 12 | 2 | $35.06 \pm 0.02$ | $2.429 \pm 0.02$ | 0.95 | 47.8M (52.0M) | $236.41 \pm 4.35$ |
| FFN Reservoir | 6 | 2 | $34.85 \pm 0.10$ | $1.729 \pm 0.03$ | 0.87 | 35.3M (37.4M) | $161.72 \pm 2.32$ |
| | 8 | 2 | $34.99 \pm 0.11$ | $1.751 \pm 0.02$ | 0.81 | 39.5M (41.6M) | $180.21 \pm 2.68$ |
| | 10 | 2 | $34.92 \pm 0.03$ | $1.907 \pm 0.02$ | 0.81 | 43.7M (45.8M) | $191.40 \pm 2.49$ |
| | 12 | 2 | $35.16 \pm 0.04$ | $2.395 \pm 0.01$ | 0.94 | 47.8M (49.9M) | $216.08 \pm 2.57$ |
| LayerDrop | 6 | 2 | $34.51 \pm 0.12$ | $1.908 \pm 0.04$ | 0.96 | 35.3M (39.5M) | $169.62 \pm 3.16$ |
| | 8 | 2 | $34.77 \pm 0.11$ | $2.023 \pm 0.02$ | 0.94 | 39.5M (43.7M) | $186.71 \pm 2.17$ |
| | 10 | 2 | $34.06 \pm 0.05$ | $1.912 \pm 0.02$ | 0.97 | 43.7M (47.9M) | $205.52 \pm 3.31$ |
| | 12 | 2 | $34.08 \pm 0.13$ | $2.524 \pm 0.01$ | 0.99 | 47.8M (52.0M) | $222.45 \pm 2.21$ |

Table 2: Wall-clock time (averaged over multiple runs) saved for IWSLT for different model types and encoder depths. Max BLEU is for validation. Number of layers is for encoder, decoder depth is kept fixed at 6. Ratio is computed compared to comparable number of layers in the normal case.

| Model | # Layers | Frozen | Max BLEU | Train time until max (in hours) | Ratio | # Params Trainable (Total) | Train Time each epoch (in hours) |
|---|---|---|---|---|---|---|---|
| Transformer | 12 | 0 | $24.46 \pm 0.04$ | $15.15 \pm 0.15$ | 1 | 75.6M | $0.505 \pm 0.005$ |
| | 16 | 0 | $24.52 \pm 0.03$ | $16.05 \pm 0.18$ | 1 | 88.2M | $0.643 \pm 0.006$ |
| | 24 | 0 | $24.69 \pm 0.05$ | $17.61 \pm 0.85$ | 1 | 113.4M | $0.877 \pm 0.029$ |
| | 32 | 0 | $24.83 \pm 0.04$ | $18.42 \pm 0.28$ | 1 | 138.6M | $1.036 \pm 0.010$ |
| T Reservoir | 12 | 4 | $24.26 \pm 0.08$ | $14.11 \pm 0.21$ | 0.93 | 72.4M (75.6M) | $0.472 \pm 0.007$ |
| | 16 | 4 | $24.50 \pm 0.05$ | $15.25 \pm 0.28$ | 0.95 | 75.6M (88.2M) | $0.596 \pm 0.009$ |
| | 24 | 4 | $25.11 \pm 0.07$ | $15.89 \pm 0.74$ | 0.90 | 100.8M (113.4M) | $0.776 \pm 0.024$ |
| | 32 | 4 | $24.66 \pm 0.04$ | $16.38 \pm 0.24$ | 0.88 | 126.0M (138.6M) | $0.998 \pm 0.009$ |
| FFN Reservoir | 12 | 4 | $24.42 \pm 0.05$ | $14.01 \pm 0.09$ | 0.92 | 72.4M (71.4M) | $0.441 \pm 0.003$ |
| | 16 | 4 | $24.65 \pm 0.07$ | $14.53 \pm 0.17$ | 0.91 | 75.6M (83.9M) | $0.524 \pm 0.006$ |
| | 24 | 4 | $24.93 \pm 0.04$ | $12.62 \pm 1.53$ | 0.71 | 100.8M (109.2M) | $0.743 \pm 0.018$ |
| | 32 | 4 | $24.98 \pm 0.03$ | $13.96 \pm 0.19$ | 0.73 | 126.0M (134.4M) | $0.964 \pm 0.007$ |
| LayerDrop | 12 | 4 | $24.27 \pm 0.03$ | $14.61 \pm 0.14$ | 0.96 | 72.4M (75.6M) | $0.489 \pm 0.006$ |
| | 16 | 4 | $24.15 \pm 0.06$ | $15.55 \pm 0.54$ | 0.97 | 75.6M (88.2M) | $0.597 \pm 0.017$ |
| | 24 | 4 | $24.37 \pm 0.05$ | $16.25 \pm 0.36$ | 0.92 | 100.8M (113.4M) | $0.823 \pm 0.013$ |
| | 32 | 4 | $23.84 \pm 0.03$ | $15.27 \pm 0.38$ | 0.83 | 126.0M (138.6M) | $1.028 \pm 0.012$ |

Table 3: Wall-clock time (averaged over multiple runs) saved for WMT for different model types and encoder depths. Max BLEU is for validation. Number of layers is for encoder, decoder depth is kept fixed at 1. Ratio is computed compared to comparable number of layers in the normal case.

# D  RoBERTa Results

Here we present the additional RoBERTa results for convergence plot and AUCC in various decoder depth setting in Figure 10. As stated in the main paper, the difference of AUCC / Convergence Plot between RoBERTa model with or without Reservoir layers are limited. Moreover, we plot the downstream task performance for SST-2 and MNLI compared to the pretraining wall-clock time in Figure 9. It can be seen that the FFN Reservoir can achieve up to 25% and 10% pretraining time savings while matching the best performance of vanilla transformers for MNLI-m and SST2, respectively.

# E  Reservoir Results for total layers

Here we present the *shifted* Reservoir Results for IWSLT14, WMT16, Enwik8 and RoBERTa fine-tuning in Figure 11, 12, 13, 14, respectively. We show the same results also hold when it comes to replace normal transformer blocks with Reservoir blocks at least for MT.

| Model | # Layers | IWSLT-Dec2 Train time until 95% max (in hours) | Max BLEU (95%) | # Layers | IWSLT-Dec6 Train time until 95% max (in hours) | Max BLEU (95%) | # Layers | WMT-Dec1 Train time until 95% max (in hours) | Max BLEU (95%) |
|---|---|---|---|---|---|---|---|---|---|
| Transformer | 6 | $0.647 \pm 0.03$ | $32.89 \pm 0.04$ | 6 | $0.642 \pm 0.02$ | $33.36 \pm 0.03$ | 12 | $3.788 \pm 0.053$ | $23.36 \pm 0.06$ |
|  | 8 | $0.711 \pm 0.05$ | $33.04 \pm 0.03$ | 8 | $0.765 \pm 0.03$ | $33.41 \pm 0.08$ | 16 | $3.820 \pm 0.072$ | $23.41 \pm 0.05$ |
|  | 10 | $0.808 \pm 0.02$ | $33.96 \pm 0.08$ | 10 | $0.898 \pm 0.04$ | $33.32 \pm 0.07$ | 24 | $5.262 \pm 0.607$ | $23.50 \pm 0.03$ |
|  | 12 | $1.037 \pm 0.03$ | $33.07 \pm 0.09$ | 12 | $1.037 \pm 0.03$ | $33.07 \pm 0.11$ | 32 | $6.212 \pm 0.232$ | $23.81 \pm 0.04$ |
| T Reservoir | 6 | $0.569 \pm 0.02$ | $32.78 \pm 0.03$ | 6 | $0.599 \pm 0.01$ | $33.09 \pm 0.05$ | 12 | $3.563 \pm 0.061$ | $23.21 \pm 0.04$ |
|  | 8 | $0.619 \pm 0.04$ | $33.12 \pm 0.05$ | 8 | $0.726 \pm 0.02$ | $33.38 \pm 0.09$ | 16 | $3.603 \pm 0.056$ | $23.80 \pm 0.06$ |
|  | 10 | $0.729 \pm 0.04$ | $33.13 \pm 0.07$ | 10 | $0.738 \pm 0.03$ | $33.37 \pm 0.04$ | 24 | $4.923 \pm 0.771$ | $23.75 \pm 0.02$ |
|  | 12 | $0.982 \pm 0.02$ | $33.03 \pm 0.11$ | 12 | $0.958 \pm 0.01$ | $33.46 \pm 0.09$ | 32 | $5.780 \pm 0.214$ | $23.71 \pm 0.03$ |
| FFN Reservoir | 6 | $0.521 \pm 0.05$ | $32.85 \pm 0.02$ | 6 | $0.594 \pm 0.03$ | $33.13 \pm 0.04$ | 12 | $3.417 \pm 0.046$ | $23.22 \pm 0.07$ |
|  | 8 | $0.533 \pm 0.03$ | $33.84 \pm 0.04$ | 8 | $0.651 \pm 0.04$ | $33.36 \pm 0.06$ | 16 | $3.527 \pm 0.063$ | $23.54 \pm 0.05$ |
|  | 10 | $0.614 \pm 0.01$ | $33.05 \pm 0.08$ | 10 | $0.627 \pm 0.05$ | $33.26 \pm 0.03$ | 24 | $4.197 \pm 0.697$ | $23.74 \pm 0.06$ |
|  | 12 | $0.811 \pm 0.02$ | $33.26 \pm 0.10$ | 12 | $0.780 \pm 0.02$ | $33.46 \pm 0.08$ | 32 | $4.984 \pm 0.321$ | $23.82 \pm 0.02$ |
| LayerDrop | 6 | $0.837 \pm 0.08$ | $32.87 \pm 0.05$ | 6 | $0.706 \pm 0.02$ | $33.08 \pm 0.03$ | 12 | $3.912 \pm 0.068$ | $23.33 \pm 0.08$ |
|  | 8 | $0.934 \pm 0.07$ | $33.12 \pm 0.03$ | 8 | $0.753 \pm 0.04$ | $33.14 \pm 0.05$ | 16 | $3.581 \pm 0.076$ | $23.17 \pm 0.04$ |
|  | 10 | $0.901 \pm 0.06$ | $33.18 \pm 0.02$ | 10 | $0.691 \pm 0.03$ | $32.39 \pm 0.05$ | 24 | $4.875 \pm 0.728$ | $23.43 \pm 0.07$ |
|  | 12 | $0.914 \pm 0.01$ | $32.33 \pm 0.06$ | 12 | $0.803 \pm 0.02$ | $32.94 \pm 0.10$ | 32 | $5.980 \pm 0.219$ | $22.97 \pm 0.08$ |

Table 4: Wall-clock time (averaged over multiple runs) saved for IWSLT/WMT for different model types and encoder depths. 95% Max BLEU is for validation.

| Model | # Layers | IWSLT-Dec2 Train time until 99% max (in hours) | Max BLEU (99%) | # Layers | IWSLT-Dec6 Train time until 99% max (in hours) | Max BLEU (99%) | # Layers | WMT-Dec1 Train time until 99% max (in hours) | Max BLEU (99%) |
|---|---|---|---|---|---|---|---|---|---|
| Transformer | 6 | $1.454 \pm 0.06$ | $34.24 \pm 0.05$ | 6 | $1.297 \pm 0.03$ | $34.69 \pm 0.05$ | 12 | $9.961 \pm 0.053$ | $24.27 \pm 0.04$ |
|  | 8 | $1.475 \pm 0.09$ | $34.32 \pm 0.09$ | 8 | $1.390 \pm 0.02$ | $34.75 \pm 0.09$ | 16 | $12.623 \pm 0.072$ | $24.35 \pm 0.06$ |
|  | 10 | $1.526 \pm 0.04$ | $34.25 \pm 0.04$ | 10 | $1.622 \pm 0.05$ | $34.64 \pm 0.03$ | 24 | $13.412 \pm 0.837$ | $24.49 \pm 0.07$ |
|  | 12 | $2.259 \pm 0.07$ | $34.24 \pm 0.11$ | 12 | $1.748 \pm 0.08$ | $34.66 \pm 0.08$ | 32 | $15.117 \pm 0.232$ | $24.56 \pm 0.02$ |
| T Reservoir | 6 | $1.257 \pm 0.04$ | $34.05 \pm 0.09$ | 6 | $1.291 \pm 0.03$ | $34.51 \pm 0.10$ | 12 | $8.314 \pm 0.062$ | $24.15 \pm 0.06$ |
|  | 8 | $1.472 \pm 0.06$ | $34.47 \pm 0.05$ | 8 | $1.339 \pm 0.03$ | $34.80 \pm 0.04$ | 16 | $9.221 \pm 0.073$ | $24.41 \pm 0.05$ |
|  | 10 | $1.530 \pm 0.03$ | $34.36 \pm 0.02$ | 10 | $1.419 \pm 0.04$ | $34.72 \pm 0.03$ | 24 | $10.413 \pm 0.580$ | $24.56 \pm 0.03$ |
|  | 12 | $2.043 \pm 0.05$ | $34.53 \pm 0.07$ | 12 | $1.642 \pm 0.02$ | $34.87 \pm 0.02$ | 32 | $11.465 \pm 0.227$ | $24.49 \pm 0.01$ |
| FFN Reservoir | 6 | $1.138 \pm 0.03$ | $34.10 \pm 0.13$ | 6 | $1.169 \pm 0.02$ | $34.71 \pm 0.09$ | 12 | $7.407 \pm 0.087$ | $24.33 \pm 0.08$ |
|  | 8 | $1.101 \pm 0.07$ | $34.32 \pm 0.11$ | 8 | $1.201 \pm 0.03$ | $34.79 \pm 0.08$ | 16 | $9.336 \pm 0.036$ | $24.42 \pm 0.05$ |
|  | 10 | $1.281 \pm 0.01$ | $34.36 \pm 0.03$ | 10 | $1.276 \pm 0.03$ | $34.63 \pm 0.03$ | 24 | $9.978 \pm 0.546$ | $24.91 \pm 0.07$ |
|  | 12 | $1.785 \pm 0.03$ | $34.42 \pm 0.06$ | 12 | $1.440 \pm 0.01$ | $34.87 \pm 0.02$ | 32 | $10.524 \pm 0.341$ | $24.96 \pm 0.01$ |
| LayerDrop | 6 | $1.363 \pm 0.05$ | $34.58 \pm 0.14$ | 6 | $1.253 \pm 0.01$ | $34.42 \pm 0.10$ | 12 | $8.372 \pm 0.059$ | $24.17 \pm 0.04$ |
|  | 8 | $1.468 \pm 0.03$ | $34.50 \pm 0.12$ | 8 | $1.244 \pm 0.04$ | $34.44 \pm 0.09$ | 16 | $9.741 \pm 0.043$ | $23.93 \pm 0.08$ |
|  | 10 | $1.678 \pm 0.04$ | $34.52 \pm 0.07$ | 10 | $1.343 \pm 0.04$ | $33.83 \pm 0.06$ | 24 | $10.145 \pm 0.628$ | $24.07 \pm 0.09$ |
|  | 12 | $2.071 \pm 0.02$ | $33.45 \pm 0.23$ | 12 | $1.423 \pm 0.02$ | $33.97 \pm 0.12$ | 32 | $10.168 \pm 0.329$ | $23.81 \pm 0.03$ |

Table 5: Wall-clock time (averaged over multiple runs) saved for IWSLT/WMT for different model types and encoder depths. 99% Max BLEU is for validation.

## F    VALIDATION PLOTS

Here we present the validation plots for training a 8-layer encoder, 2-layer decoder model for IWSLT14, a 24-layer encoder, 1-layer decoder model for WMT14, a 48-layer decoder model for enwik8 and a 12-layer decoder model for RoBERTa for detailed steps to calculate the AUCC. It can be clearly observed that given the configurations from Section 3.1, all the models have converged. So when we compute the area under the convergence curve, this depicts the training efficiency of the model (basically time x performance) until convergence. Specifically, we set T sufficiently high for computing the AUCC, which is 4h for IWSLT, 20h for WMT, 30h for enwik8 and 60h for RoBERTa pretraning. From the training plot in the appendix, we can see that each model has converged at that point. The Reservoir model in Figure 15 has 2 layers frozen for IWSLT14, 8 layers frozen for enwik8, and 4 layers frozen for WMT14 and RoBERTa.

## G    ROBERTA PROBING

We follow Jawahar et al. (2019) and investigate what the frozen layers in the Reservoir Transformer have actually "learned" (while being forzen) as measured by probing tasks, reported in Table 6. The results are gathered over 3 random seeds for reporting the mean and standard deviation. From the table, we can see that generally probing performance is quite similar between Transformer and the T Reservoir model. We also noticed that the representations collected after the frozen layer (3, 5, 7, 9) in the T Reservoir actually have significantly better performance over the regular Transformer representations across all the probing tasks. This has interesting repercussions for the study of "BERTology", as it clearly shows, somewhat confusingly, that even completely random and frozen layers represent linguistic phenomena.

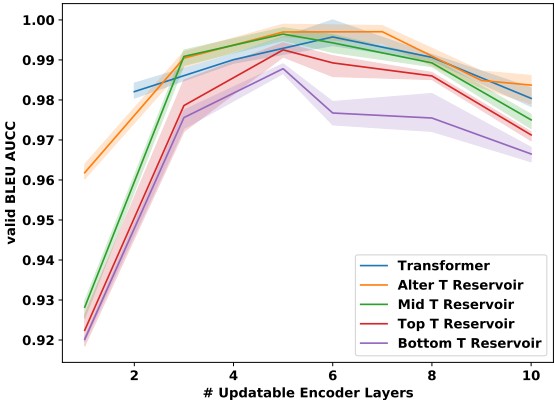

Figure 8: IWSLT with 2-layer decoder using different freezing strategy.

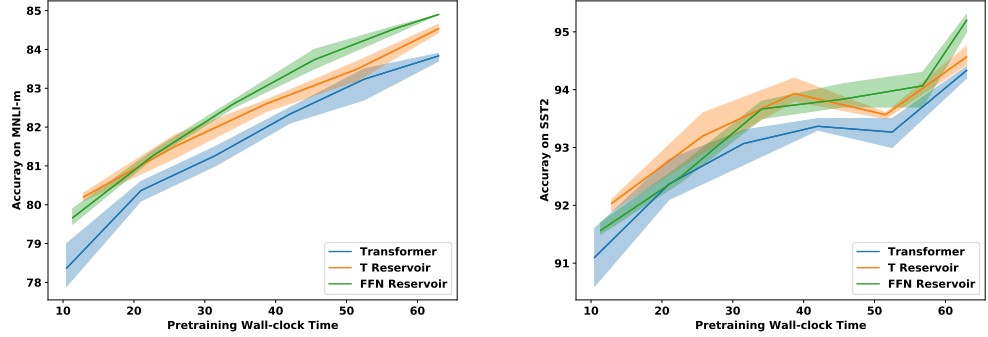

Figure 9: RoBERTa Reservoir Results, Pre-training versus downstream task plot for 12 layer RoBERTa. MNLI-m (left). SST-2 (right).

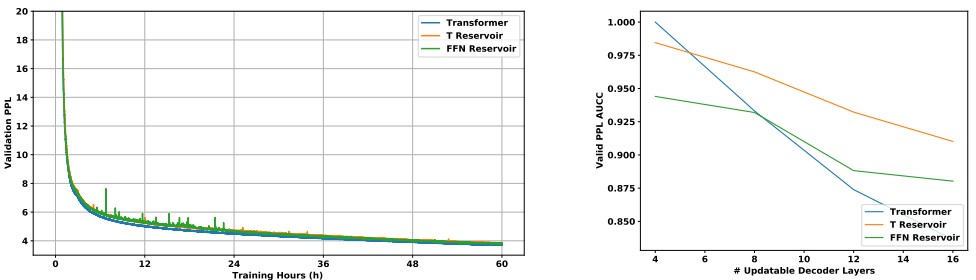

Figure 10: RoBERTa Reservoir Results, Training plot for 12 layer RoBERTa (left). AUCC result (right).

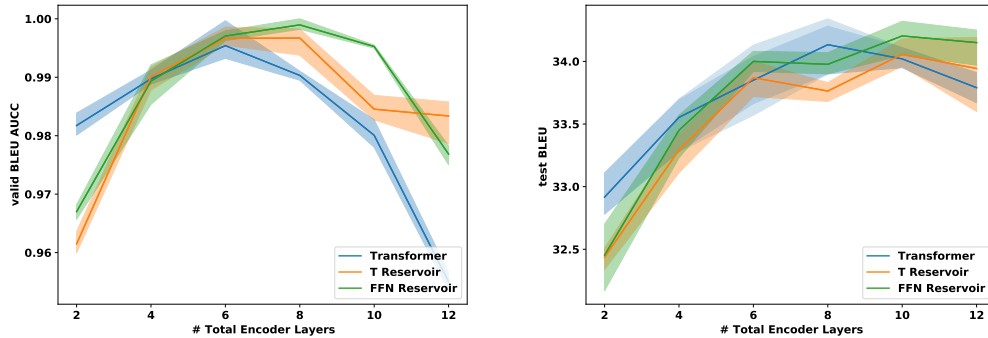

Figure 11: Validation BLEU AUCC and test BLEU for IWSLT (high is good). Comparison of regular transformer and reservoir transformer with FFN or Transformer reservoir layers added.

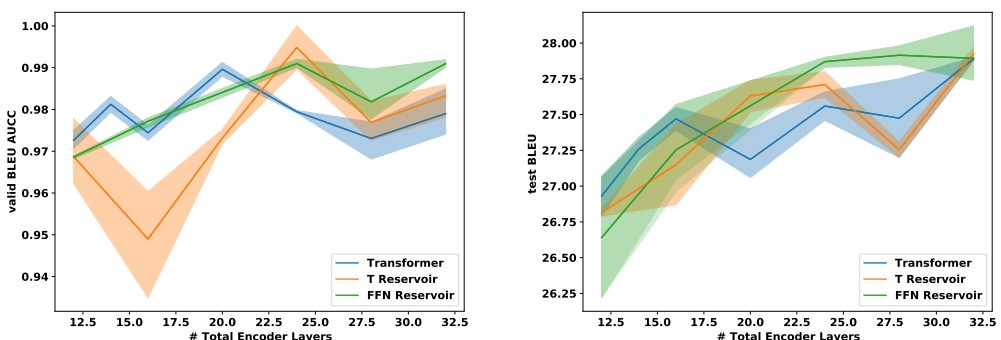

Figure 12: Validation BLEU AUCC and test BLEU for WMT (high is good). Comparison of regular transformer and reservoir transformer with FFN or Transformer reservoir layers added.

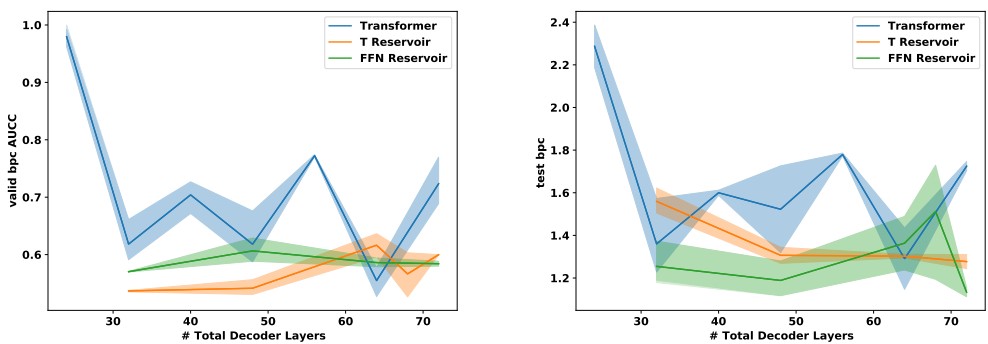

Figure 13: Validation BPC AUCC and test BPC on the enwik8 language modelling task (low is good). Comparison of regular and reservoir transformers for varying depths.

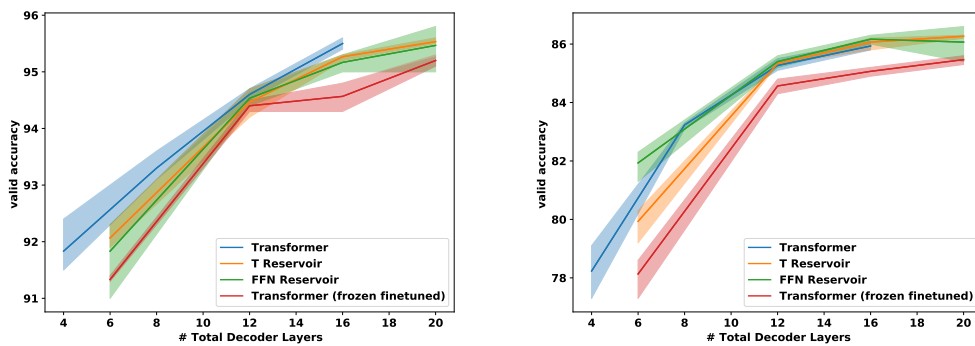

Figure 14: Downstream RoBERTa performance on SST-2 (left) and MultiNLI-matched (right).

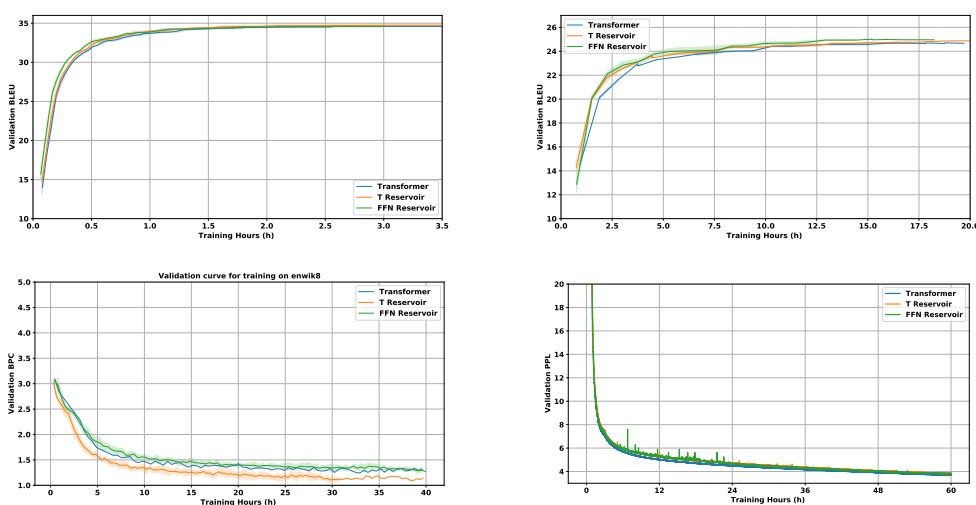

Figure 15: IWSLT with 2-layer decoder validation plot (upper left). WMT with 24-layer decoder validation plot (upper right). Enwik8 with 48-layer decoder validation plot (lower left). RoBERTa with 12-layer decoder validation plot (lower right).

| Model | Layer | SentLen (Surface) | TreeDepth (Syntactic) | TopConst (Syntactic) | BShift (Syntactic) | Tense (Semantic) | SubjNum (Semantic) | ObjNum (Semantic) | SOMO (Semantic) | CoordInv (Semantic) |
|---|---|---|---|---|---|---|---|---|---|---|
| | 1 | 84.56 ± 0.54 | 32.30 ± 0.41 | 54.40 ± 0.33 | 49.99 ± 0.01 | 80.98 ± 0.32 | 76.26 ± 0.09 | 50.01 ± 0.19 | 76.38 ± 0.61 | 54.33 ± 0.47 |
| | 2 | 87.22 ± 0.07 | 33.63 ± 0.57 | 58.38 ± 0.20 | 50.12 ± 0.17 | 82.84 ± 0.68 | 78.65 ± 0.19 | 51.47 ± 0.53 | 78.00 ± 1.12 | 54.66 ± 0.55 |
| | 3 | 84.25 ± 0.16 | 32.60 ± 0.17 | 54.41 ± 0.10 | 50.02 ± 0.01 | 81.72 ± 0.59 | 77.00 ± 0.13 | 51.32 ± 0.64 | 76.57 ± 1.13 | 54.13 ± 0.51 |
| | 4 | 87.37 ± 0.20 | 32.59 ± 0.29 | 50.06 ± 0.21 | 69.76 ± 0.26 | 81.63 ± 1.17 | 76.47 ± 0.09 | 52.41 ± 1.49 | 76.15 ± 0.84 | 52.62 ± 1.34 |
| | 5 | 84.61 ± 0.24 | 31.14 ± 0.48 | 44.76 ± 0.38 | 74.82 ± 0.11 | 80.16 ± 0.19 | 73.66 ± 0.16 | 52.95 ± 1.77 | 72.90 ± 0.21 | 51.26 ± 1.14 |
| Transformer | 6 | 82.56 ± 0.25 | 30.31 ± 0.40 | 39.30 ± 0.40 | 78.80 ± 0.38 | 81.88 ± 0.47 | 75.30 ± 0.07 | 56.21 ± 1.26 | 74.37 ± 0.16 | 51.44 ± 1.04 |
| | 7 | 70.85 ± 0.13 | 26.65 ± 0.72 | 40.70 ± 0.13 | 78.98 ± 0.32 | 85.11 ± 0.31 | 72.03 ± 0.46 | 58.15 ± 0.46 | 68.71 ± 0.91 | 55.39 ± 0.27 |
| | 8 | 66.23 ± 1.33 | 23.46 ± 0.44 | 25.19 ± 1.02 | 77.42 ± 0.27 | 80.35 ± 0.45 | 67.55 ± 0.99 | 54.94 ± 2.04 | 63.69 ± 2.32 | 50.58 ± 0.83 |
| | 9 | 71.17 ± 0.29 | 31.21 ± 0.31 | 58.42 ± 0.29 | 85.55 ± 0.44 | 86.77 ± 0.19 | 80.30 ± 0.08 | 62.01 ± 0.59 | 81.68 ± 0.45 | 66.90 ± 0.49 |
| | 10 | 73.19 ± 0.50 | 27.74 ± 0.53 | 41.01 ± 0.22 | 83.56 ± 0.96 | 86.13 ± 0.35 | 83.04 ± 0.04 | 62.01 ± 0.59 | 79.73 ± 0.21 | 62.60 ± 1.04 |
| | 11 | 71.37 ± 0.42 | 30.22 ± 0.28 | 48.58 ± 0.35 | 84.40 ± 0.44 | 87.28 ± 0.59 | 82.34 ± 0.15 | 61.10 ± 0.14 | 80.00 ± 0.40 | 64.44 ± 0.38 |
| | 12 | 71.66 ± 0.12 | 33.43 ± 0.18 | 64.38 ± 0.20 | 87.38 ± 0.02 | 88.41 ± 0.09 | 84.46 ± 0.25 | 63.01 ± 0.05 | 81.80 ± 0.27 | 65.72 ± 0.16 |
| | 1 | 87.75 ± 0.10 | 31.60 ± 0.21 | 50.38 ± 0.23 | 50.00 ± 0.00 | 80.40 ± 0.18 | 76.47 ± 0.20 | 50.53 ± 0.14 | 73.48 ± 0.15 | 53.55 ± 0.70 |
| | 2 | 81.28 ± 0.23 | 34.20 ± 0.41 | 61.41 ± 0.42 | 60.64 ± 0.65 | 81.50 ± 0.77 | 76.33 ± 0.08 | 50.73 ± 0.34 | 74.28 ± 0.67 | 56.82 ± 0.10 |
| | **3** | **89.28 ± 0.09** | **36.42 ± 0.11** | **67.36 ± 0.45** | **75.64 ± 0.52** | **85.42 ± 0.18** | **80.53 ± 0.02** | **52.50 ± 1.80** | **78.47 ± 1.81** | **57.16 ± 0.27** |
| | 4 | 74.31 ± 0.32 | 32.42 ± 0.83 | 55.19 ± 0.33 | 73.41 ± 0.00 | 79.56 ± 0.00 | 75.15 ± 0.08 | 53.68 ± 0.66 | 75.02 ± 0.19 | 56.89 ± 0.08 |
| | **5** | **88.03 ± 0.22** | **38.34 ± 0.64** | **68.65 ± 0.29** | **82.25 ± 0.12** | **86.80 ± 0.02** | **82.27 ± 0.33** | **57.95 ± 0.24** | **80.82 ± 0.91** | **58.05 ± 0.10** |
| T Reservoir | 6 | 74.55 ± 0.37 | 33.13 ± 0.29 | 52.70 ± 0.81 | 79.21 ± 0.13 | 85.70 ± 0.36 | 77.43 ± 0.03 | 57.26 ± 0.19 | 75.38 ± 0.66 | 51.95 ± 1.30 |
| | **7** | **85.82 ± 0.37** | **37.63 ± 0.13** | **70.43 ± 0.05** | **84.12 ± 0.35** | **86.88 ± 0.07** | **82.86 ± 0.30** | **61.17 ± 0.21** | **80.79 ± 0.17** | **61.83 ± 0.95** |
| | 8 | 71.69 ± 0.71 | 30.32 ± 0.01 | 48.44 ± 0.30 | 79.12 ± 0.12 | 84.75 ± 0.09 | 79.23 ± 0.11 | 59.53 ± 0.16 | 76.80 ± 0.41 | 57.34 ± 0.14 |
| | **9** | **85.86 ± 0.12** | **37.89 ± 0.03** | **69.53 ± 0.37** | **85.55 ± 0.12** | **87.98 ± 0.22** | **84.13 ± 0.01** | **63.06 ± 0.01** | **82.55 ± 0.31** | **66.07 ± 0.05** |
| | 10 | 69.22 ± 0.23 | 25.58 ± 0.35 | 29.20 ± 0.58 | 78.57 ± 0.09 | 85.02 ± 0.03 | 75.68 ± 0.16 | 57.55 ± 1.57 | 74.70 ± 0.02 | 55.02 ± 0.64 |
| | 11 | 65.70 ± 0.05 | 30.57 ± 0.03 | 47.56 ± 0.02 | 81.20 ± 0.00 | 86.78 ± 0.02 | 83.73 ± 0.05 | 60.38 ± 0.17 | 80.59 ± 0.15 | 62.50 ± 0.11 |
| | 12 | 70.61 ± 0.18 | 34.45 ± 0.20 | 64.19 ± 0.10 | 84.53 ± 0.03 | 87.48 ± 0.16 | 84.86 ± 0.14 | 62.75 ± 0.14 | 82.08 ± 0.03 | 64.73 ± 0.06 |

Table 6: RoBERTa Probing Results. The line in bold text are the the frozen layers in the T Reservoir.

