# OpenReview forum: "Reservoir Transformers"
_ICLR.cc/2021/Conference — Reject_

### Official Review · AnonReviewer2 · 2020-10-27
**Interesting and simple idea but the experiments need to be clearer**

**Rating:** 5
**Confidence:** 4

**Review:**

### Summary

The paper studies how transformers train when some layers are kept fixed at the randomly initialized parameters. The authors observe that transformers can be effectively trained with a significant percentage of their layers "frozen". It is also argued that as a result transformers can be trained more efficiently as for the frozen layers gradients with respect to the parameters need not be computed.

### Strengths

- The proposed idea is very simple and easy to understand and implement.
- The authors perform thorough experiments on several datasets and tasks.

### Weaknesses

1. The AUCC metric is not measuring the efficiency independent of the time budget as it is claimed. For instance given a high enough T_hat the best performing model will always have the highest AUCC. Contrary to that, given a low enough T_hat the fastest to converge model will always be better. I agree with the authors that measuring the efficiency of a neural network is a hard problem but I don't think that the AUCC metric is a good solution. Time to X% of best score is probably much more informative. In addition, time to best score is also not very informative as it depends heavily on the learning rate schedule and the random oscillations of training.
2. From the provided comparison with respect to the number of updateable layers we cannot deduce information regarding the efficiency. For instance in Fig. 2 for WMT how many "frozen" layers are used for the reservoir transformers? What is the wall-clock time per epoch for each model?
3. From the comparison in the supplementary material with respect to all the layers in the model we observe that the performance is on par with standard transformers but what is the improvement in efficiency? Namely, how much faster are the reservoir transformers for a given number of layers?

### Reasons for recommendation

The paper proposes a very simple idea and then evaluates it experimentally. Therefore, the experimental section should be thorough and should lead to clear conclusions. Due to the the newly proposed metric as well as the lack of comparison between training time and achieved performance lead me to propose rejection.

### Miscellaneous

- Figure 14 bottom right should probably read "Validation PPL".

### Post-rebuttal update

I would like to thank the authors for their additions to the paper. I believe that the extra metrics improve the reader's ability to extract conclusions from the experiments significantly.

Having said that, I believe that the extra experiments and numbers do not paint a clearer picture. For instance, in IWSLT and WMT indeed there is fairly consistent evidence that FFN Reservoir is more efficient to train than fully trainable transformers. However, for enwik8 the T Reservoir outperforms everything significantly when looking at figure 15 but judging from figure 13 we see that the best case scenario has been selected for T Reservoir (namely 32 or 48 layers). Even more importantly perhaps, the story is completely different when looking at the test set evaluation where FFN Reservoirs perform better and actually the T Reservoir performs the worst among all methods. Similarly on RoBERTa pretraining, fully trainable transformers seem to achieve the lowest validation perplexity and with the highest efficiency.

To summarize, I believe that the reservoir transformers could be a useful tool for improving either the efficiency or the generalization of transformers in low-data regimes or both. However, a distillation of the experiments and conclusions is required in order for this to be shown from the paper. Namely, even with all those numbers I still cannot judge which reservoir layer will be better, in what sense it will be better (accuracy or efficiency) and why is it going to be better.

Due to the above, I tend to keep my score but because the additions of the authors provide significantly more information (and in my opinion value) to the paper, I will increase my score to 5.

---

> ### Author Response · Authors · 2020-11-23
> **Paper updated with your feedback, thanks!**
>
> Thank you very much for your constructive comments! We have worked very hard to address your concerns and to incorporate your feedback in the new revision. We hope you will find the new version more to your liking, and hope you can revise your scores accordingly. In detail:
>
> Thanks for the constructive comments on the AUCC metric. As you say, measuring the efficiency-accuracy trade-off is not a trivial problem. As we say in the paper (Section 3), “One potential downside of this approach is that the AUCC metric could lead to higher scores for a model that converges quickly but to ultimately worse performance, if measured in a small window. We account for this by making sure that $\hat{T}$ is set sufficiently high [4h for IWSLT, 20h for WMT, 40h for enwik8 and 60h for RoBERTa].” For the same reason, we included the raw validation curves in the appendix and also report test set generalization in each experiment, so that anyone can check for themselves that our efficiency-v-performance claims hold up. From the training plots in the appendix, one can see that each model has converged long before the point where we set T.
>
> We do want to make sure that we’ve carefully addressed your concerns and agree that AUCC is not perfect: hence, we have added additional metrics, as per your suggestions, to demonstrate the differences in model efficiency and performance. Specifically, we added your suggested “Time to X% of best score” metric to Tables 1, 2 and 3; report the number of parameters; added wall-clock times per your request, and included standard deviations everywhere to ensure random oscillations cannot cause discrepancies. This also makes it easier to see how much faster the reservoir transformers are for a given number of layers. The speedup gain as measured by “time to best” is up to 27% by using a FFN Reservoir. We also added time to 95% and 99% results in Tables 4 and 5 in the appendix. We'd be happy to provide multiple different percentages in the final version, if you like.
>
> We provide the detailed training curves in the Appendix. We added the pretraining wall-clock time to the RoBERTa finetuning performance plots in the Appendix as well. Compared to the vanilla 12 layer Transformer model, the FFN Reservoir consistently outperforms the vanilla model in each time-step. By matching the best performance with the vanilla model, the FFN Reservoir can save up to 25% pretraining wall-clock time for MNLI-m and 10% pretraining wall-clock time for SST2.
>
> We added the wall-clock time per epoch for each model and added a strong LayerDrop baseline per R5’s request. It can be clearly seen that even though LayerDrop performs similarly in terms of efficiency with T Reservoir at an average 3% to 7% saving on wall-clock time per epoch (and as networks grow deeper, this difference grows bigger), FFN Reservoir consistently outperforms LayerDrop at average 5%~15% saving on wall-clock time per epoch.
>
> We also explicitly state how many "frozen" layers are used for the reservoir transformers for each task now. For the number of frozen layers, we provide the “un-shifted” version of the AUCC plots in the appendix, which shows the total number of layers in each setting for easier comparison. We have clarified this in the new revision.
>
> Your suggestions led us to think about what sorts of things individual layers, whether frozen or not, actually learn. In Appendix G we report results for probing individual RoBERTa layers in the Transformer and T Reservoir case, shedding interesting new light on the question of what individual layers learn from a computational linguistics perspective. Very surprisingly, it appears that some reservoir layers, without “learning”, do much better on some of these probes than their learned counterparts.
>
> We really appreciate your constructive feedback - it has helped us make the paper much better, and we really hope we’ve sufficiently addressed your concerns for you to update your score to accept.

---

### Official Review · AnonReviewer1 · 2020-10-29
**randomisation in transformer networks**

**Rating:** 7
**Confidence:** 4

**Review:**

The paper explores the concept of randomization in Transformer architectures. Essentially, some of the layers in the encoder network are replaced by fixed layers, which makes the computation faster straightforwardly in the prediction phase. The paper also illustrates a method, called backskipping, to reduce the cost of backpropagating gradients through the fixed layers. The proposal is well supported by a number of experiments in the area of Machine Translation.
PROs:
- I find the paper very clear and well written & readable. I enjoyed reading it.
- The research proposed in the paper seems novel enough to me, and goes towards a fruitful research direction, that one of randomized neural algorithms (although this is nowadays established in the ML community).
- The experiments offer a nice perspective on the advantages of the proposed model

CONs:
- I find inappropriate the use of the term "reservoir" in this work. In my understanding, that is essentially a fixed (untrained) recurrent layer. The aspect of the dynamics is fundamental in this respect. In the paper, instead, the use of "reservoir" stands mainly for "randomized". All in all, I don't see reservoir layers in the proposed model, and I suggest to take this into account (perhaps changing the name and the title)
- The experiments show that using untrained layers gives a positive trade-off between accuracy and compute times. In the experimental comparison, I think it would - however - important to see how this trade-off behaves/scales wrt to the number of trainable weights (when avoiding training in some of the layers the number of trainable weights is reduced, and the overall complexity of the system is reduced too, having a sort of regularization effect).
- The experiments show that it is possible to avoid training in some of the layers of a Transformer. However, in randomized neural networks, a pivotal role is played by the scaling parameters of the involved weight matrices. Orthogonal matrices are used in the experiments, which is good, but I wonder how (and if) the scaling of such matrices affect the performance. Ideally such scaling should be hyper-parameters and be chosen on a validation set.

a few minor points:
- please, if possible, clarify more explicitly on the concept of "converging faster"
- please, clarify on the data splitting (in tr/vl/ts) for the used datasets
- please, introduce the datasets before section 3.1 (where they are already mentioned without intro).

-- EDIT:
I would like to thank the authors for the nice work during the review process. I am pretty satisfied with that and I feel serene to increase my rating to acceptance.

---

> ### Author Response · Authors · 2020-11-23
> **Paper updated with your feedback, thanks!**
>
> Thanks for your super valuable feedback! We’re very glad that you enjoyed reading our paper and that you think the results are novel and interesting. We have revised the paper, incorporating your feedback as best as we could in this short timeframe. In particular, we really liked your suggestion to show the number of trainable and non-trainable parameters, and have added this to Tables 1, 2 and 3. We also did preliminary experiments to examine the impact of scaling on initializations, and added clarifications for what we mean by “converging faster” and what datasets (including exact splits) we use in the work. We hope that you can revise your rating accordingly given this new information. In more detail:
>
> 1. Regarding the meaning of “reservoir”: You make a valid point that the term reservoir is often employed in the context of RNNs. In fact, we had initially titled our work “Echo State Transformers” but decided against this because the approach does not necessarily abide by the echo state property. Since Lukosevicius & Jaeger in their review refer to a “reservoir” as something somewhat more general, which is “randomly created and remains unchanged during training”, we take this to be compatible with our highly non-linear random transformations in transformers, especially given the close connections between transformers and RNNs (just to name one relevant paper: “Transformers are RNNs”, Katharopoulos et al., ICML 2020). That said, if you think that this is not appropriate we would be more than happy to change the name.
>
> 2. Number of parameters: you make an excellent point, we added this important information to the paper in the result tables, showing both the number of total parameters as well as the number of trainable parameters. In Table 1, 2 and 3 respectively, it can be seen that our results generalize from 39.5M (IWSLT), 75.6M (WMT) to 153.0M (RoBERTa).
>
> 3. Thanks for the suggestion to explore the impact of the scaling parameters. We did some preliminary experiments, where we either a) scale by 1 as normal, b) scale by the global norm of that layer from a pre-trained same-sized network, c) scale by the local norm of that module from a pre-trained same-sized network on the validation set. What find that 1 performs the best on IWSLT with an average 34.6 Validation BLEU across different depths of the transformer. Using the global / local norm will significantly hurt the performance to around an average 25.8 Validation BLEU across different depths of the transformer. We will conduct more thorough experiments in the final version as an ablation study of the sensitivity of this scaling factor.
>
> 4. Clarification about “converging faster”: Per R2’s request, we’ve also added the time to (% of) the best validation score for the results tables. Converging faster can be interpreted there as achieving better/similar validation performance in a shorter amount of training time. For the FFN Reservoir, we achieve up to 27% convergence gain for a 24 layer Transformer model, which is significantly better than vanilla Transformer or the newly added LayerDrop baseline results.
>
> 5. We also provide additional RoBERTa pretraining results when fine-tuning on downstream tasks in the Appendix in Figure 9. Each model has the same number of updatable layers (12) there. From the figure, we can clearly see that for the MLM-m task, both FFN Reservoir and T Reservoir constantly achieve better performance in each wall-clock time-stamp. Moreover, to match the best performed accuracy of the vanilla Transformer, we can save up to 25% of the pretraining time by using the FFN Reservoir.
>
> 6. Data splitting for the used datasets: For WMT, we follow the pre-processing steps the same as Vaswani et al. (2017). The train/val/test split is 4.5M/16.5k/3k sentences. For IWSLT, we follow the pre-processing steps in Edunov et al. (2018). The train/val/test split is 129k/10k/6.8k sentences. For enwik8, we follow the pre-processing steps in Dai et al. (2019). The train/val/test split is 1M/54k/56k sentences. For RoBERTa pretraining, we follow the pre-processing steps in Liu et al. (2019). We added these details to Section 3.1, thanks for the great suggestion.

---

### Official Review · AnonReviewer5 · 2020-11-04
**Review #5**

**Rating:** 5
**Confidence:** 3

**Review:**

The authors demonstrate that transformers obtain impressive performance even when some of the layers are randomly initialized and never updated. The authors have experiments on four types of reservoir: Transformer Reservoir , FFN (feed-forward layer) Reservoir, BiGRU Reservoir and CNN Reservoir. And the results show that the Reservoir can achieve competitive/better performance or than normal Transformer on machine translation, language modeling, and MLM pre-training. As pointed out by the authors,  deep reservoir computing networks (Scardapane & Wang, 2017; Gallicchio & Micheli, 2017) have been explored before. The main novelty of this paper is only the Reservoir exploration of Transformer structure. Although the method is interesting and the experiments are well-done, the work "REDUCING TRANSFORMER DEPTH ON DEMAND WITH STRUCTURED DROPOUT" seems more straightforward and effective. The authors should have some comparisons with this method. As there's still residual connections, how about simply adding some noise to the previous layer? Overall, although the experiments are interesting, it doesn't provide novel theory on it and misses some comparison to previous works. It's hard to evaluate the significance of this work.

Pro:
1. The reservoir on Transformer is interesting and has not been explored before.
2. The authors prove the idea by many different tasks.

Cons:
1. The reservoir operation was explored on other structures.
2. No strong baseline provided, such as "REDUCING TRANSFORMER DEPTH ON DEMAND WITH STRUCTURED DROPOUT" or some other structure pruning based methods.
3. No novel theory to explain the method.


######update

I like the experiment added in the revision. However, it is only tested on a IWSLT which is a smaller dataset and can be influence by many hyper-parameters. It's not clear what frozen layers mean for LayerDrop and it need to be clarified with more details. I didn't find clear comparison with stronger baselines on WMT in the revision.

As pointed our by the authors, I think the theory mainly come from previous works.

I have also read other reviews. Overall, I would like to keep my rating.

---

> ### Author Response · Authors · 2020-11-23
> **Paper updated with your feedback, thanks!**
>
> We thank you for your valuable comments! We have updated the paper, incorporating your feedback as best we could. In particular, we have added results for your requested baseline and show that our method performs similarly or better in the T reservoir case, and outperforms it in the FFN case. We really appreciate your suggestions, and we hope that you can revise your rating accordingly given this new information.
>
> At your request, we performed a series of additional experiments with LayerDrop (Fan et al., “Reducing Transformer Depth on Demand with Structured Dropout”) as a baseline. We have updated Tables 1, 2, 3, 4 and 5 with this new information, ensuring that the number of updatable layers is the same and results are comparable. We run all experiments with three random seeds to obtain error bounds. As shown, we can see T Reservoir achieves similar performance as LayerDrop on IWSLT where LayerDrop in terms of wall-clock per epoch and wall-clock time to the best performance. On WMT, where the network needs to be deeper, LayerDrop turns out to achieve worse efficiency gains compared to the T Reservoir. Moreover, on both tasks, FFN Reservoir turns out to perform much better than LayerDrop in terms of efficiency per epoch and achieves better/similar performance in less amount of time in each case. We would be happy to also do this experiment for the language modelling and pre-training settings in the final version of the paper, if you so desire.
>
> You make a good point about the theoretical underpinnings of this work. We agree that there is lots of room for further exploration of this phenomenon, and it has interesting relationships with e.g. the lottery ticket hypothesis. However, as we argue in the paper (Section 2.1), this result is actually reasonably well-established from a theory perspective in the ML literature, with Rahimi & Recht (2008) being the most recent theoretical treatment of this phenomenon. We definitely hope that our empirical findings will inspire more thorough theoretical treatments of this phenomenon.

---

### Decision · Program_Chairs · 2021-01-07
**Final Decision**

**Decision:**

Reject

**Comment:**

The reviewers were split between accept (7) and borderline reject (two 5's). All three reviewers acknowledged that the proposed approach is simple and intuitive (but this paper follows, for the most part, the concept of reservoir operation and apply it to transformers). The main criticisms were insufficient experiments (R5) and the lack of a clear conclusion (R2). I found these concerns to be valid and did not find strong reasons to overturn their recommendations. More comprehensive experiments (especially on WMT) and clear conclusions (accuracy or efficiency) would make this paper much stronger.